# Nanoparticles of Mixed-Valence Oxides Mn_X_CO_3-X_O_4_ (0 ≤ X ≤ 1) Obtained with Agar-Agar from Red Algae (Rhodophyta) for Oxygen Evolution Reaction

**DOI:** 10.3390/nano12183170

**Published:** 2022-09-13

**Authors:** Jakeline Raiane D. Santos, Rafael A. Raimundo, Thayse R. Silva, Vinícius D. Silva, Daniel A. Macedo, Francisco J. A. Loureiro, Marco A. M. Torres, Domenica Tonelli, Uílame U. Gomes

**Affiliations:** 1Materials Science and Engineering Postgraduate Program, UFRN, Natal 59078-970, Brazil; 2Department of Physics, UFPB, João Pessoa 58051-900, Brazil; 3Materials Science and Engineering Postgraduate Program, UFPB, João Pessoa 58051-900, Brazil; 4Centre for Mechanical Technology and Automation, Mechanical Engineering Department, UA, 3810-193 Aveiro, Portugal; 5Department of Industrial Chemistry “Toso Montanari”, Industrial Chemistry, UNIBO, V.le Risorgimento 4, 40136 Bologna, Italy

**Keywords:** electrocatalyst, green synthesis, proteic sol-gel, mixed-valence oxides

## Abstract

The development of efficient electrocatalysts for the oxygen evolution reaction (OER) is of paramount importance in sustainable water-splitting technology for hydrogen production. In this context, this work reports mixed-valence oxide samples of the Mn_X_Co_3-X_O_4_ type (0 ≤ X ≤ 1) synthesized for the first time by the proteic sol-gel method using Agar-Agar as a polymerizing agent. The powders were calcined at 1173 K, characterized by FESEM, XRD, RAMAN, UV–Vis, FT-IR, VSM, and XPS analyses, and were investigated as electrocatalysts for the oxygen evolution reaction (OER). Through XRD analysis, it was observed that the pure cubic phase was obtained for all samples. The presence of Co^3+^, Co^2+^, Mn^2+^, Mn^3+^, and Mn^4+^ was confirmed by X-ray spectroscopy (XPS). Regarding the magnetic measurements, a paramagnetic behavior at 300 K was observed for all samples. As far as OER is concerned, it was investigated in an alkaline medium, where the best overpotential of 299 mV vs. RHE was observed for the sample (MnCo_2_O_4_), which is a lower value than those of noble metal electrocatalysts in the literature, together with a Tafel slope of 52 mV dec^−1^, and excellent electrochemical stability for 15 h. Therefore, the green synthesis method presented in this work showed great potential for obtaining electrocatalysts used in the oxygen evolution reaction for water splitting.

## 1. Introduction

In recent decades, with the exponential growth of the population, the intensification of using fossil fuels has generated great impacts on the environment [1]. Given this, the energy transition from fossil fuels to clean energy sources has become necessary, and major renewable energy technologies have been developed [2], such as geothermal [3], wind [4], solar [5,6], and biomass [7]. However, to benefit resources even more, it is necessary to use efficient devices for energy storage and conversion [8,9].

One of the energy conversion processes that has received a lot of attention is water splitting via electrolysis for the production of hydrogen [10], which consists of two semi-reactions: the hydrogen evolution reaction (HER) [11] and the oxygen evolution reaction (OER) [12,13,14]. During the water-splitting process, the kinetic reaction that takes place at the anode (OER) is slow and requires high energy consumption (overpotential) due to the four electrons transferred in the reaction, causing its efficiency to decrease [15]. Therefore, the use of effective electrocatalysts is necessary to accelerate the reaction and reduce the overpotential [16]. However, the most widely used and effective electrocatalysts for water splitting are compounds based on noble metals such as ruthenium and iridium, but the high cost and scarcity of these elements have limited their large-scale application [17,18]. Therefore, one of the main challenges is to develop new electrocatalysts made of low-cost and abundant materials while also offering high electrochemical performance [19].

In this context, mixed-valence transition metal oxides such as the Mn_X_Co_3-X_O_4_ (Mn-Co-O) system have attracted attention, as they represent an important class of multifunctional materials for electrocatalysis [20], oxygen evolution reaction [21], oxygen reduction reaction [22], fuel cells [23], and batteries [24]. These applications are related to the intrinsic properties of Mn-Co-O due to the multiple oxidation states of the metal ions [25]. This material has a spinel-like structure, with general formula A[B]_2_O_4_, where A is represented by a cation in the tetrahedral site (Coordination Number 4), and B is a cation in the octahedral site (Coordination Number 6) [26]. Depending on the occupancy of the divalent (A) and trivalent (B) cations, spinels are classified into three categories, normal, inverse, and complex [27], which gives them interesting physical, chemical [28], and electrochemical properties [29].

Furthermore, the distribution of cations between the different coordination sites strongly depends on the synthesis conditions [30]; thus, several syntheses have been developed in recent years to obtain mixed-valence transition metal oxides. Among the most common methods for obtaining (Mn-Co-O) are the sol-gel [31], combustion [32], solvothermal [33], solid-state [34], co-precipitation [35], spray pyrolysis [36], and hydrothermal [37] methods. These processes directly influence the morphology and/or structure of the materials, especially the sol-gel method, which allows crystalline materials to be obtained [38,39].

However, another method that has been attracting the attention of many researchers is that named proteic sol-gel synthesis, which is defined as a modification of the traditional sol-gel method, and allows ceramic materials to be obtained at the nanometer scale [40]. This method involves the following steps: preparation of a composite solution, formation of an amorphous network via polymerization reactions, followed by hydrolysis, gel formation, removal of organic residues and water, and finally calcination [41,42]. In this context, the proteic sol-gel method has several advantages over the traditional method, such as simplicity, low cost, higher speed, and generation of less waste to the environment [43,44], since it employs organic precursors, which are called polymerizing agents, having the function of replacing citric acid and ethylene glycol as driving agents of the reaction. Furthermore, these agents have hydroxyl and carboxyl groups that promote the chelation of metal ions during the synthesis process [45]. Some examples of organic precursors are flavorless gelatin [46], coconut water [47], and Agar-Agar [48].

Agar-Agar is a biopolymer [49], hydrocolloid [50] and polysaccharide source [51], extracted from red algae of the class Rhodophyceae [52], which is composed of two main parts: agarose, which is responsible for the gelling ability, and agaropectin, which is the polymeric part resulting from the existence of various substituent groups, such as sulfates and methyl ethers among others [53]. Due to its gelling properties, biocompatibility, biodegradability, and non-toxicity [54], Agar-Agar is used in the food, leather, cosmetic, beverage, and pharmaceutical industries [55].

The present work aims to investigate the structural, optical, and magnetic properties through the Co/Mn ratios in the structure, as well as the electrochemical properties of Mn_X_Co_3-X_O_4_-type mixed-valence transition metal oxide powders synthesized by the proteic sol-gel method using Agar-Agar as a polymerizing agent.

## 2. Experimental Section

### 2.1. Materials

Cobalt nitrate (Co(NO_3_)_2_·6H_2_O, (Sigma-Aldrich 99%, Saint-Louis, MO, USA), manganese nitrate (Mn(NO_3_)_2_·4H_2_O, Vetec 99%, Saint-Louis, MO, USA), and Agar-Agar (Gelialgas-Agargel, João Pessoa, Brazil) were used in the present study. Nickel foam (Ni 99.8%, porosity >95%) was purchased from QiJing Ltd., Ninghai, China.

### 2.2. Synthesis of the Proteic Sol-Gel Using Agar-Agar

The scheme of Mn_X_Co_3-X_O_4_ (0 ≤ X ≤ 1) preparation is shown in Figure 1. The mixed oxides were synthesized using Co(NO_3_)_2_ 6H_2_O and Mn(NO_3_)_2_ 4H_2_O, while Agar-Agar was used as the polymerizing agent. First, 2.0 g of Agar-Agar was dispersed in 50 mL of distilled water at 60 °C, then the proper amounts in mols of the metal salts were added as follows for samples with X = 0.0 (cobalt nitrate: 0.0249 mol), X = 0.2 (cobalt nitrate: 0.0233 mol and manganese nitrate: 1.6673 mmol), X = 0.4 (cobalt nitrate: 0.0217 mol and manganese nitrate: 3.3437 mmol), X = 0.6 (cobalt nitrate: 0.0201 mol and manganese nitrate: 5.0337 mmol), X = 0.8 (cobalt nitrate: 0.0185 mol and manganese nitrate: 6.7332 mmol), and X = 1.0 (cobalt nitrate: 0.0169 mol and manganese nitrate: 8.4459 mmol), and the resulting solution was kept under stirring at 90 °C until the formation of a gel. The resulting gel was kept at 350 °C for 2 h. Hence, the obtained powders were ground and calcined in air at 900 °C. The samples Mn_X_Co_3-X_O_4_ were labeled as: for X = 0.0 (Co_3_O_4_), X = 0.2 (Mn_0.2_Co_2.8_O_4_), X = 0.4 (Mn_0.4_Co_2.6_O_4_), X = 0.6 (Mn_0.6_Co_2.4_O_4_), X = 0.8 (Mn_0.8_Co_2.2_O_4_) and X = 1.0 (MnCo_2_O_4_).

### 2.3. Structural and Morphological Characterization

X-ray powder diffraction patterns (XRD) were obtained by a Shimadzu XRD-7000 diffractometer using Kα(Cu) = 1.5481 Å radiation. The 2θ range was investigated from 10° to 80° with a step size of 0.02° and acquisition time of 1 s. The crystallite size, lattice parameters, and atomic positions were determined by Rietveld refinement using the software Materials Analysis Using Diffraction (TOPAS). FT-IR spectra were performed by a Shimadzu IRPrestige21 spectrophotometer between 500 and 4000 cm^−1^, using KBr pellets. The ultraviolet–visible spectra (UV–Vis) were obtained in the UV-2600i spectrophotometer from Shimadzu. SEM images were obtained by a field-emission scanning electron microscope (FESEM, Carl Zeiss, Supra 35-VP Model) equipped with a Bruker EDS detector (XFlash 410-M). Surface chemical states were studied by X-ray photoelectron spectroscopy (XPS) using a SPECS Phoibos 150 spectrometer with a high-intensity monochromatic Al-Ka X-ray source (1486.6 eV). Samples were dispersed in acetone and deposited on silicon by drop-coating. Adventitious carbon C 1s with binding energy at 284.8 eV was used as reference energy. CasaXPS software was used for spectra deconvolution, thus obtaining the height, area, and position of the analyzed peaks. All the symmetric peaks were fitted using Gaussian and Lorentzian functions. Magnetic measurements were obtained using a vibrating sample magnetometer (VSM) from Lakeshore, model 7400, at room temperature, with a maximum magnetic field applied up to +15.0 KOe.

### 2.4. Electrochemical Characterization

All electrochemical studies were performed in an alkaline aqueous solution (KOH, 1 M pH = 13.6) at room temperature by a PGSTAT204 with FRA32M module (Metrohm Autolab) using a three-electrode setup with a platinum plate and Hg/HgO as counter and reference electrodes, respectively. The samples Mn_X_Co_3-X_O_4_ (0 ≤ X ≤ 1) were used for the fabrication of the working electrodes. Catalytic inks were prepared by mixing 5 mg of each catalyst with 50 μL of Nafion solution (5 wt%) and dispersing the mixture in 500 μL of isopropyl alcohol. Then, inks were drop-casted onto Ni foams (1 × 1 cm) on clean substrates and dried at room temperature for 5 h to prepare the working electrodes. Linear sweep voltammetry (LSV) was performed at 5 mV s^−1^. Electrochemical impedance spectroscopy (EIS) was carried out using dc potentials (1.4 V vs. RHE) in the frequency range of 0.1 Hz–10 kHz and voltage amplitude of 10 mV. All measured potentials (with iR correction) were converted to the reversible hydrogen electrode (RHE) using the Nernst equation (E_RHE_ = E_Hg/HgO_ + 0.059×pH + 0.098). Overpotential (ղ) values were calculated by the equation ղ = E_RHE_—1.23 V. The stability tests were conducted by chronopotentiometry analysis using multi-steps of 10–20 mA cm^−2^. Figure 2 shows the complete characterization process.

## 3. Results

### 3.1. X-ray Diffraction (XRD)

The refined X-ray diffraction patterns of the Mn_X_Co_3-X_O_4_ (0 ≤ X ≤ 1) are shown in Figure 3a. As noted, all peaks are characteristic of the cubic phases of Co_3_O_4_ (structure of Spinel#MgAl_2_O_4_ type, with lattice parameter *a* = b = 8.072(3) Å, ICSD n° 36256, space group Fd-3mS (227)) [56] and MnCo_2_O_4_ (structure of Spinel#MgAl_2_O_4_ type, with lattice parameter *a* = b = 8.28(2) Å, ICSD n° 18544, space group Fd-3mZ (227)) [57]. No secondary phases corresponding to impurities were detected. The ICSD n° 36,256 was used to fit the Mn_X_Co_3-X_O_4_ samples (X < 1), while the 18,544 ICSD file was applied to refine the sample with composition X = 1. The observed patterns are similar to those reported previously for pure and doped cobaltites [16,20,58]. All crystallographic parameters, including crystallite size and lattice parameter, as well as the agreement indices (*R_wp_*, *R_exp_*
*e* χ^2^) for samples of Mn_X_Co_3-X_O_4_ (0 ≤ X ≤ 1) are gathered in Table 1.

Figure 3b shows the magnification of the most intense diffraction peak (311), located between 36° and 37.5°. As shown, increasing Mn content shifts the (311) peak to lower angles, indicating a continuous increase in the lattice parameter from 8.0757(1) for x = 0, 8.0759(8) for x = 0.2, 8.1183(7) for x = 0.6, 8.1675(8) for x = 0.8 to 8.2381(6) for x = 1. The total width at half maximum intensity (FWHM) also increases gradually with the increase in Mn (0.1322 for x = 0, 0.1433 for x = 0.2, 0.1513 for x = 0.4, 0.1613 for x = 0.6, 0.1633 for x = 0.8 and 0.1692 for x = 1), signaling a progressive reduction in crystallite size and increase in strain (90.6 nm for x = 0, 82.5 nm for x = 0.2, 80 nm for x = 0.4, 71 nm for x = 0.6, 68.8 nm for x = 0.8, 66 nm for x = 1, with the only exception of the sample corresponding to X = 0.4 for which the 2θ angle is slightly higher than the one displayed by the sample with X = 0.2). The largest variations of FWHM occur in the 0 ≤ X ≤ 0.6 range. The Mn^+2^ has a radius of 0.80 Å and it is larger than the radii of Co^+3^ (0.63 Å) and Co^+2^ (0.65 Å); thus, it causes distortion and strain in the Co_3_O_4_ lattice, resulting in a decrease in crystallite size. In addition, other manganese oxidation states may be present in the samples such as Mn^+3^ and Mn^+4^. Mn^+3^ (3d^4^) is responsible for the Jahn–Teller phenomenon, which also develops a distortion in the lattice and an intrinsic strain that leads to a decrease in the crystallite size [20,59,60,61]. The trends of lattice parameter and crystallite size are in agreement with previous reports [16,20]. The maximum values of the agreement factors *R_wp_* and *R_exp_* from the Rietveld analyses are 8.14% and 6.92%, respectively. The low values of fitting quality (χ^2^ ≤ 1.30) indicate excellent agreement between the data and the refined models.

### 3.2. Field-Emission Scanning Electron Microscopy (FESEM)

The FESEM images of the Mn_X_Co_3-X_O_4_ nanoparticles (0 ≤ X ≤ 1) are shown in Figure 4. A non-uniform morphology was observed, specifically polyhedral-shaped particles, and a few smaller spherical-like particles, mostly agglomerated [38]. Another observation is that as the amount of Mn increases, the morphology of particles tends to be octahedral like. The average particle size distribution was 208 nm for X = 0.0, 162 nm for X = 0.2, 145 nm for X = 0.4, 142 nm for X = 0.6, 140 nm for X = 0.8, and 133 nm for X = 1.0. From the size distribution histograms, it is evident the shift of main sizes to smaller values as the Mn content increased.

### 3.3. Transmission Electron Microscopy (TEM)

Additional morphological characterization was carried out by the TEM technique. Typical images of the nanoparticles are presented in Figure 5. They show particles with non-uniform morphologies of different sizes. These pictures agree well with the images acquired by FESEM (Figure 4). Figure 5b,e,h,k,n,q shows high-resolution TEM images (5 nm scale) of particles larger than 10 nm in size, with fringes related to atomic planes with spacings of 0.24, 0.296, 0.282, 0.252, 0.303, and 0.486 nm that may be due to the planes (311), (220), (220), (311), (220) and (111) for samples X = 0.0, X = 0.2, X = 0.4, X = 0.6, X = 0.8 and X = 1.0, respectively. Furthermore, it appears that the particles are coated with a carbon layer with a thickness smaller than 5 nm. Figure 5c,f,i,l,o,r shows the small-area electron diffraction (SAED) patterns of the samples. They exhibit diffraction rings originating from crystal planes (111), (220), (311), (400), (422), (333), and (440). The planes are listed beginning from the smallest ring.

### 3.4. Fourier-Transform Infrared (FT-IR) Spectroscopy

The FT-IR technique shows the vibrational fingerprint of the sample, with absorption peaks that correspond to the frequencies of vibrations of the bonds among the atoms that make up the material [62]. Figure 6 shows the spectra of the Mn_X_Co_3-X_O_4_ samples (0 ≤ X ≤ 1) in the range from 400 to 4000 cm^−1^, where two bands with the highest intensities are located at 552–570 and 643–663 cm^−1^, which are related to the stretching vibrations of the metal–oxygen bond, which confirms the formation of the pure Co_3_O_4_ phase [63]. The *v*_1_ band at 552–570 cm^−1^ is characteristic of the vibration of Co^3+^ at the octahedral site, and the *ν*_2_ band at 643–663 cm^−1^ is related to the vibration of Co^2+^ at the tetrahedral site, confirming the formation of the spinel-like oxide [64], in agreement with the XRD study. The low-intensity band that appears at 1100 cm^−1^ is due to C-O stretching vibrations. The band at 1383 cm^−1^ is attributed to the symmetric deformations of C-N and CH_2_ groups, originating from the residues of nitrate ions and agar-agar [65,66,67]. The band at 1635 cm^−1^ was attributed to the angular deformation of the adsorbed water molecules [68]. The broad absorption band in the region of about 3440 cm^−1^ is due to OH stretching of the water molecules adsorbed from the moisture during the storage process [69]. Furthermore, it is observed that the bands at 554–643 cm^−1^ for sample X = 0.8 and the bands at 552–643 cm^−1^ for sample X = 1.0 are similar, which may be related to the oxidation states of manganese (Mn^+2^, Mn^+3^, and Mn^+4^) or to the vibrational intensity between the manganese and oxygen bond. In general, the frequency of the peaks of the absorption bands at (552–570) and (643–663) cm^−1^ (Table 2) decreases with the replacement of cobalt with manganese ions, i.e., they shift to the right as the amount of manganese increases, and this is related to the increase in the metal–oxygen distance, as indicated by the increase in the lattice parameter of the unit cells (Table 1), since Mn ions are larger than Co ions [21].

### 3.5. Ultraviolet–Visible Spectroscopy (UV–Vis)

The electronic properties of Mn_X_Co_3-X_O_4_ samples, as illustrated in Figure 7, were investigated by UV–Vis spectroscopy in the wavelength range from 300 to 1400 nm. The absorptions at 528 and 792 nm for the sample X = 0 correspond to the ligand–metal O(-II) → Co(III) and O(-II) → Co(II) electron transfer, respectively [76,77,78]. The variation of absorption in the range from 1033 to 1110 nm shows that when the amount of manganese increases, the absorption peak wavelength increases, and this is related to the higher O_2_/O_1_ ratios according to the XPS results, and it will affect the overpotential in the oxygen evolution reaction for the MnCo_2_O_4_ (X = 1.0) sample, which would mean a better catalytic activity for the oxidation reactions.

### 3.6. Raman Spectroscopy

Figure 8 shows the Raman spectra of the Mn_X_Co_3-X_O_4_ samples in the range from 100 to 1000 cm^−1^. The observed bands in the intervals 193–688 cm^−1^ for x = 0.0, 186–682 cm^−1^ for x = 0.2, 186–675 cm^−1^ for x = 0.4, 185–667 cm^−1^ for x = 0.6, 183–661 cm^−1^ for x = 0.8, and 182–660 cm^−1^ for x = 1.0 correspond to the active Raman modes A1g+Eg+3F2g (Table 3), confirming the formation of the pure phase of mixed-valence oxides of spinel-like structure [79,80,81,82,83]. The most intense band at 688–660 cm^−1^ is assigned to the octahedral site MO_6_ related to the A1g mode of the O_7h_ spectroscopic symmetry, which corresponds to the stretching vibrational modes of these oxides M-O, where M = {Co, Mn}, thus substantiating the formation of MnCo_2_O_4_. The Raman bands with medium intensity in the intervals 468–488 cm^−1^ and 508–518 cm^−1^ are assigned to Eg and F2g, respectively; meanwhile, the Raman bands with lower intensities in the interval 603–617 cm^−1^ are caused by the F2g mode. Moreover, the Raman bands with very low intensity at 182–193 cm^−1^ are attributed to the F2g mode related to the tetrahedral sites of CoO_4_ [82,83,84]. In general, when comparing the positions of the peaks, it is observed that as the amount of manganese increases, the peaks shift to the left, analogously to what was noticed in the FT-IR spectra. This change may be due to the greater ionic radius of Mn^2+^, in comparison to that of Co^2+^/Co^3+^ [85,86,87], which when entering the structure of Co_3_O_4_, generates a large distortion in the crystalline structure and increases the distance between the metal and the oxygen, and consequently a weakening of bonds occurs. Another reason would be due to vibrations in the structure, where the Co^2+^ and Co^3+^ cations are located in tetrahedral and octahedral sites in the cubic crystal structure [88].

### 3.7. X-ray Photoelectron Spectroscopy (XPS)

The surface oxidation states of the samples were analyzed by XPS. Figure 9 shows the high-resolution Co 2p, Mn 2p, and O 1s spectra obtained from the analysis. All data were corrected for the carbon peak position. In the case of the Co 2p spectra (Figure 9a), four peaks were deconvoluted, at lower binding energies, corresponding to Co^3+^ and Co^2+^, as well as two satellite peaks at higher binding energies. The binding energies obtained for Co^3+^ were found to fall in the 779.64–780.19 eV range, while for Co^2+^, in the 781.19–781.85 eV range, in agreement with previous work [90]. We also found a Co^2+^/Co^3+^ ratio varying from 0.31 to 0.40 among the samples, for which the sample X = 0.8 was found to have the lowest value of Co^2+^/Co^3+^ = 0.31. Higher oxidation states can induce more bonded oxygen species, which may have a positive impact on oxidation reactions.

Conversely, in the case of the Mn 2p spectra (Figure 9b), the data were deconvoluted into three peaks, which were ascribed to Mn^4+^ (ranging from 644.154 eV to 645.146 eV), Mn^3+^ (ranging from 642.790 eV to 643.125 eV), and Mn^2+^ (641.223 eV to 641.459 eV), in agreement with previous reports [90,91]. The lowest oxidation state species, Mn^2+^, was found to represent the largest fraction of the total species present at the surface, with a relative value varying from 0.42 to 0.51 among the samples. Conversely, the presence of Mn^4+^ and Mn^3+^ oxidation states is related to the relatively high calcination temperature used in this work, i.e., 900 °C, as found in previous literature [92]. In this respect, we also noted a higher Mn^3+^/Mn^4+^ ratio for the tested samples, which correlates well with the previously discussed Jahn–Teller phenomenon, with a concurrent distortion of the crystal lattice.

Finally, the O 1s high-resolution spectra (Figure 9c) display three deconvoluted peaks: O_1_ (529.94–530.27 eV), O_2_ (531.16–531.72 eV), and O_3_ (532.55–533.19 eV). Based on the characteristic binding energies determined for these peaks, they are likely related to surface lattice oxygen (O_lat_, O^2−^), adsorbed oxygen species (O_ads_, O^2−^, O_2_^2−^, and O^−^), and adsorbed water species (O_H2O_), in agreement with earlier reports on similar compounds [91,93]. From the analysis of the O 1s high-resolution spectra (Table 4), we determined higher O_2_/O_1_ ratios with increasing Mn content, with a maximum value obtained for the X = 1.0 sample. This suggests that the compounds with the highest Mn content possess increased catalytic activity, as a likely result of increased oxygen-ion vacancies in these samples.

### 3.8. Vibrating Sample Magnetometer (VSM)

Magnetization measurements were done to study the magnetic behavior of samples at room temperature and to determine the cation magnetic moment in an approximate manner. For all samples, the isothermal magnetization at T = 300 K showed a linear behavior with the magnetic field, and their magnetization at a given field increased with the Mn concentration, as shown in Figure 10.

This trend was observed for all samples and it is typical of paramagnetic samples as shown in Figure 10. From the classical theory of paramagnetism we know that the relationship between the magnetization (M) and magnetic field (H) is given by the Langevin function L(*a*) = M/M_o_ = coth(*a)*-1/*a*, where M and M_o_ are the mass magnetizations per total amount of Mn and Co (without oxygen), *a* = μH/K_B_T, μ is the average magnetic moment per cation, K_B_ is the Boltzmann constant (1.3807 × 10^−16^ cm^2^gK/s^2^), and T = 300 K. It is known that L(*a*) tends to *a*/3 when *a* is less than about 0.5 [94]. In the present case, if μ~4.51 × 10^−20^ Erg/Oe (theoretical magnetic moment for Mn^3+^ and Co^3+^) and H = 15 × 10^3^ Oe, then, one can get *a* = 0.01633, which is smaller than 0.5. Thus, L(*a*) = M/M_o_ ≈ *a*/3 and, therefore, M = [M_o_μ/(3K_B_T)]H. In a paramagnetic system M_o_ = Nμ/*A*, where N is the Avogadro’s number and *A* = (x*54.938 + (3-x)*58.933)/3 is the average atomic mass provided by Mn and Co in Mn_X_Co_3-X_O_4_ (where X = {0.0, 0.2, 0.4, 0.6, 0.8, 1.0}). Thus, the DC susceptibility is χ = M/H= Nμ^2^/(3*A*K_B_T) [94]. One can study the M-H data by fitting the curve to a linear equation and comparing the slope to Nμ^2^/(3*A*K_B_T). Then, to obtain μ in Bohr magnetons (μ_B_), one has to calculate μ/0.9274 × 10^−20^. The results provided an effective magnetic moment per cation of 2.204 2.294, 2.342, 2.344, 2.348, 2.378 μ_B_ for the samples prepared with x = 0.0, 0.2, 0.4, 0.6, 0.8, 1.0, respectively.

The magnetic moments expected for low spin configuration have a total spin of *S*(Mn^2+^) = 0.5, *S*(Mn^3+^) = 0, *S*(Mn^4+^) = 0.5, *S*(Co^2+^) = 0.5, *S*(Co^3+^) = 0, whose magnetic moments are, μ = $2\sqrt{S\left(S+1\right)}$ μ_B_, i.e., μ(Mn^2+^) = 1.732 μ_B_, μ(Mn^3+^) = 0, μ(Mn^4+^) = 1.732 μ_B_, μ(Co^2+^) = 1.732 μ_B_, μ(Co^3+^) = 0. Furthermore, for high spin configuration *S*(Mn^2+^) = 2.5, *S*(Mn^3+^) = 2.0, *S*(Mn^4+^) = 1.5, *S*(Co^2+^) = 1.5, *S*(Co^3+^) = 2, whose magnetic moments are μ(Mn^2+^) = 5.916 μ_B_, μ(Mn^3+^) = 4.899 μ_B_, μ(Mn^4+^) = 3.873 μ_B_, μ(Co^2+^) = 3.873 μ_B_, μ(Co^3+^) = 4.899 μ_B_. Therefore, the magnetic moments for Co and Mn seem to be mainly in the low spin configuration; however, we cannot rule out the presence of some moments in the high spin configuration.

### 3.9. Oxygen Evolution Reaction (OER)

The samples were also evaluated as electrocatalysts for the oxygen evolution reaction (OER). According to the results of the anodic polarization (Figure 11a), the electrodes presented values of 515 (Ni foam), 342 (X = 0.0), 342 (X = 0.2), 339 (X = 0.4), 337 (X = 0.6), 323 (X = 0.8), 299 (X = 1.0) and 235 (RuO_2_/Ni foam benchmark, extracted from reference [95]) mV vs. RHE, respectively, to record a current density J = 10 mA cm^2^. Among the investigated materials, the MnCo_2_O_4_ (X = 1.0) samples displayed the best catalytic activity for OER, i.e., the lowest overpotential because the incorporation of Mn into the structure enhanced the defect concentrations, thus increasing the amount of catalytically active sites, which facilitated the mass transfer process, favoring OER [96]. Moreover, the crystalline size decreased with the increase of manganese content, which indicates that the Co_3_O_4_ sample (X = 0.0) has larger average crystal sizes than the other samples, especially MnCo_2_O_4_ (X = 1.0); thus, Co_3_O_4_ was the sample that had the highest overpotential. This indicates that the presence of Mn, has a suppressive effect on Co_3_O_4_ [97]. The obtained values are in agreement with others reported in the literature for Mn_X_Co_3-X_O_4_ nanostructures, as shown in Table 5.

The electrocatalytic kinetics for OER was investigated by the Tafel plots extracted from the LSV (linear sweep voltammetry) curves (Figure 11a), using the Tafel equation (η = a + b log j), where b is the Tafel slope, η is the overpotential, j is the current density, and a is a constant. The values of the Tafel slope (Figure 11b) were 63 (X = 0.0), 73 (X = 0.2), 72 (X = 0.4), 69 (X = 0.6), 68 (X = 0.8), and 52 mV dec^−1^ (X = 1.0). Therefore, the results did not follow exactly a sequence like the ղ_10_ values (Figure 11a), but it can be observed that the electrode based on the MnCo_2_O_4_ sample (X = 1.0) exhibited the best reaction kinetics for OER, as it showed the lowest Tafel slope, which demonstrates a higher efficiency for oxygen evolution. The Tafel slope of 63 mV dec^−1^ for the Co_3_O_4_ sample (X = 0.0) corresponds to slightly slower kinetics, indicating limitation in charge and mass transfer processes compared to the x = 1.0 sample. The Co_3_O_4_ sample was also the one with the highest overpotential, with no distortion in the structure, which reduces defects and consequently the oxygen vacancies [106,107]. These results are consistent with the XPS values as well as the electrochemical impedance spectroscopy. The samples with X = 0.2, 0.4, 0.6, and 0.8 show values next to 70 mV dec^−1^, which means much slower kinetics for OER.

All this evidence can be explained by the distortion of the lattice with the increase of the amount of Mn in the Co_3_O_4_ structure, which changes the electronic charge distribution and increases the disorder in the crystalline system [106,107,108,109]. Furthermore, with increasing Mn percentages, the availability of oxygen and flexibility in the lattice is greater, which in turn is related to the M-O bond length [110]. In any case, the samples (X = 0.2), (X = 0.4), (X = 0.6) and (X = 0.8) show values of Tafel slope below 80 mV dec^−1^, and these results indicate the adsorption of intermediate species as the rate-determining step (rds), based on the Krasil’shchikov reaction model for OER in alkaline medium [111,112].

The double-layer capacitance (C_DL_) can be obtained from the relationship between the anode current density (i_a_) and the scan rate (ʋ), according to (i_a_ = ʋ x C_DL_) [100]. Figure 11c shows the double-layer capacitance values obtained for the samples: 1.78 (X = 0.0), 2.75 (X = 0.2), 2.57 (X = 0.4), 2.41 (X = 0.6), 3.16 (X = 0.8), and 2.03 mF cm^−2^ (X = 1.0). These results suggest that the largest number of active sites is organized on the electrode surfaces. Although among the samples of this series, the one with X = 0.8 had the second lowest performance for OER, it displays the highest C_DL_ value, which may be linked to the high amount of oxygen vacancies that improves the absorption of reactive species (like OH¯) [113]. For the samples (X = 0.2), (X = 0.4), and (X = 0.6), the values of 2.75, 2.57, and 2.41 mF cm^−2^, respectively, are consistent with the XPS data (Figure 9b), where the species in the lowest oxidation state, Mn^2+^, represented the largest fraction of the total species present on the surface, with a relative fraction ranging from 0.42–0.51 among the samples. However, even with a low C_DL_ value for the sample (X = 1.0), the presence of Mn ions in the structure is essential for superior electrocatalytic properties. This was proven by the best overpotential extracted from the LSV curves (Figure 11a) and the XPS data (Figure 9c), as the substitution of Mn in spinel oxide cobalt occurs selectively in the (Co^3+^) lattices, and the energy required for Mn^2+^ to substitute Co^3+^ is lower than that of Co^2+^, [114]. Moreover, Mn^+2^, Mn^+3^, and Mn^+4^ have ionic ratios of 0.80, 0.66, and 0.60 Å, respectively, and the ratios of Mn^+2^ and Mn^+3^ are larger than that of Co^3+^ (0.63 Å). Therefore, Mn doping results in the expansion of the Co_3_O_4_ lattice, generating defects, which influences the mass diffusion and charge transfer properties, contributing to oxygen-ion vacancies, which are consistent with the O_2_/O_1_ ratio that was highest for the MnCo_2_O_4_ sample (X = 1.0), with a value of 0.940 [108,109,115,116].

Durability is another important indicator of catalytic performance. The stability of the electrocatalysts was evaluated by chronopotentiometry. Tests were performed at a current density of 10 mA cm^−2^ for 15 h. According to the curves shown in Figure 11d, it can be seen that the samples (X = 0) and (X = 0.6) exhibited a potential decrease until about 2 h, but then they remained stable, whereas for the samples (X = 0.8) and (X = 1.0), the potential was practically stable for the entire time period tested. In general, all samples showed satisfactory stability over 15 h of testing [117].

### 3.10. Electrochemical Impedance Spectroscopy (EIS)

The electrocatalytic activity also was investigated by electrochemical impedance spectroscopy (EIS). The EIS spectra of all samples were collected at 1.4 V vs. RHE. As seen in the Bode plots (Figure 12b), the OER is composed of complex processes involving electrosorption of intermediate species during the reaction progress. This suggests an equivalent circuit model (ECM) able to describe these processes [118,119] that is composed of R_S_ (uncompensated solution resistance), R_P_ (polarization resistance, which denotes the overall rate of the OER), *Q*_DL_ (double-layer pseudo-capacitance), R_-ad_ (resistance associated with adsorption of intermediate species), and *Q*_-ad_ (pseudo-capacitance of these species throughout the reaction). A constant phase element (*Q*) was used to model an imperfect capacitor, and its impedance was obtained by:(1)$$Z_{Q}=(Q{\left(i\omega {)}^{n}\right)}^{-1}$$

Then, the values were used to calculate true capacitance (C_DL_ or C_-ad_) by:(2)$$C=R^{\left(1-n\right)/n}\;Q^{1/n}$$

In Figure 12a, the impedance of the electrodes is composed of two incomplete semicircles. The first is attributed to the polarization process (charge transfer), and the second indicates limitations on mass transfer processes, related to the intermediate species adsorption process [120]. For the electrodes, the obtained R_P_ values were consistent with the OER performance, i.e., the X = 1.0 sample showed the lowest value (6.02 Ω), followed by X = 0.8 (9.70 Ω). The other samples revealed R_P_ values very close, but the result was expected as their overpotential values were close. The C_DL_ values varied slightly (Table 6) due to the oxidation peak shown in Figure 11a. The R_ad_C_ad_ loop associated with relaxation, which was attributed to the adsorbed intermediate species, revealed the difficulty of these electrodes to work in the diffusive processes observed at low frequencies. The high R_-ad_ (>1100 Ω) values displayed by the electrodes in those low frequency (>1 Hz) confirm that the adsorption of intermediates should be a rate-limiting step as predicted by Tafel analysis (Figure 11b) [118,121]. The values obtained from the fitting of the spectra are listed in Table 6.

## 4. Conclusions

The Mn_X_Co_3-X_O_4_ (0 ≤ X ≤ 1) samples were synthesized by the proteic sol-gel method (green synthesis) using Agar-Agar as a polymerizing agent in order to investigate their structural, optical, magnetic, and electrochemical properties. X-ray diffraction indicated for all samples the obtainment of the pure cubic phase without any secondary phase, which was also confirmed from Raman, TEM, FT-IR, and UV–Vis studies. Regarding the magnetic measurements, it was observed for all samples a magnetization in a certain field increasing with the Mn concentration, which is typical of a paramagnetic behavior. From the XPS analysis, the species in the Mn^2+^ oxidation state represented the largest fraction of the total species present on the surface, and as the amount of Mn increased, the O_2_/O_1_ ratio also increased, reaching a value of 0.940 for the sample MnCo_2_O_4_ (X = 1.0). For OER, the same sample exhibited the best catalytic activity when compared with the others, with an overpotential of 299 mV, which is lower than those of noble metal electrocatalysts reported in the literature. In addition, the samples showed superior long-term stability for efficient water oxidation activities at J = 10 mA/cm^2^ per 15 h. Thus, it can be concluded that proteic sol-gel synthesis is an excellent method to produce nanosized mixed-valence oxides Mn_X_Co_3-X_O_4_ for the fabrication of electrodes for water electrolysis.

## Figures and Tables

**Figure 1 nanomaterials-12-03170-f001:**
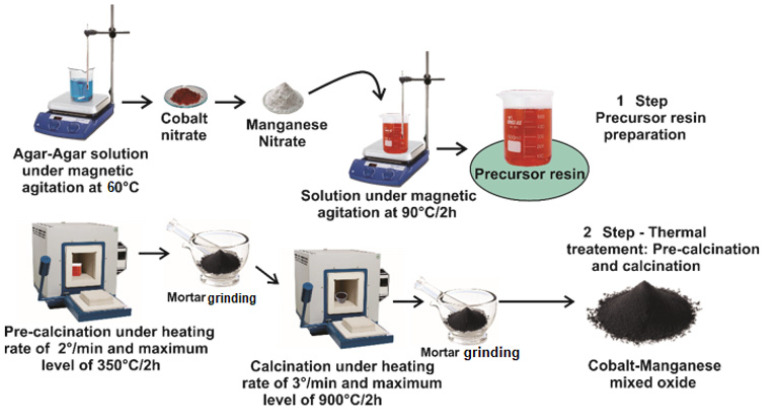
Procedure for the synthesis of samples Mn_X_Co_3-X_O_4_ (0 ≤ X ≤ 1).

**Figure 2 nanomaterials-12-03170-f002:**
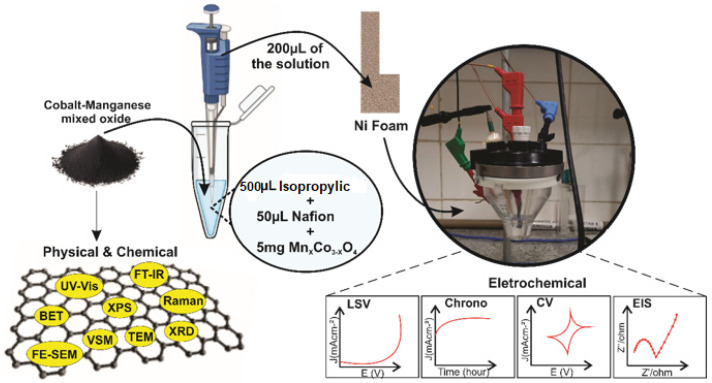
Physical, chemical, and electrochemical characterization of Mn_X_Co_3-X_O_4_ (0 ≤ X ≤ 1).

**Figure 3 nanomaterials-12-03170-f003:**
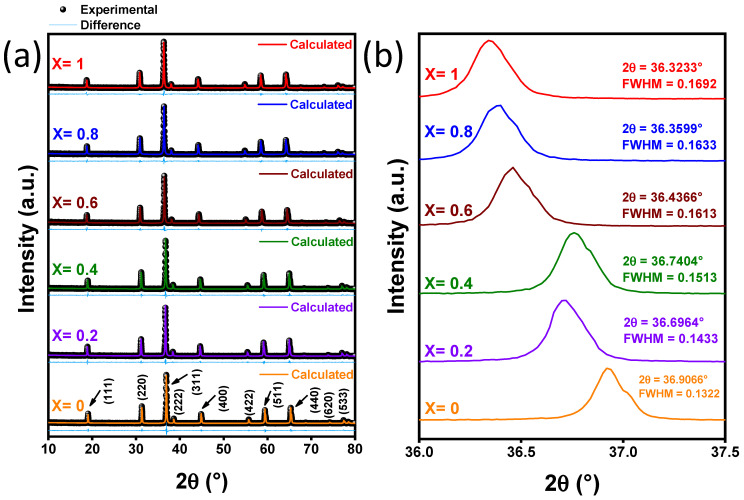
(**a**) XRD patterns of Mn_X_Co_3-X_O_4_ (0 ≤ X ≤ 1) samples (**b**) Peaks (311) for each sample and their positions and FWHM. Blue lines below the diffractograms are the difference between the calculated and experimental data.

**Figure 4 nanomaterials-12-03170-f004:**
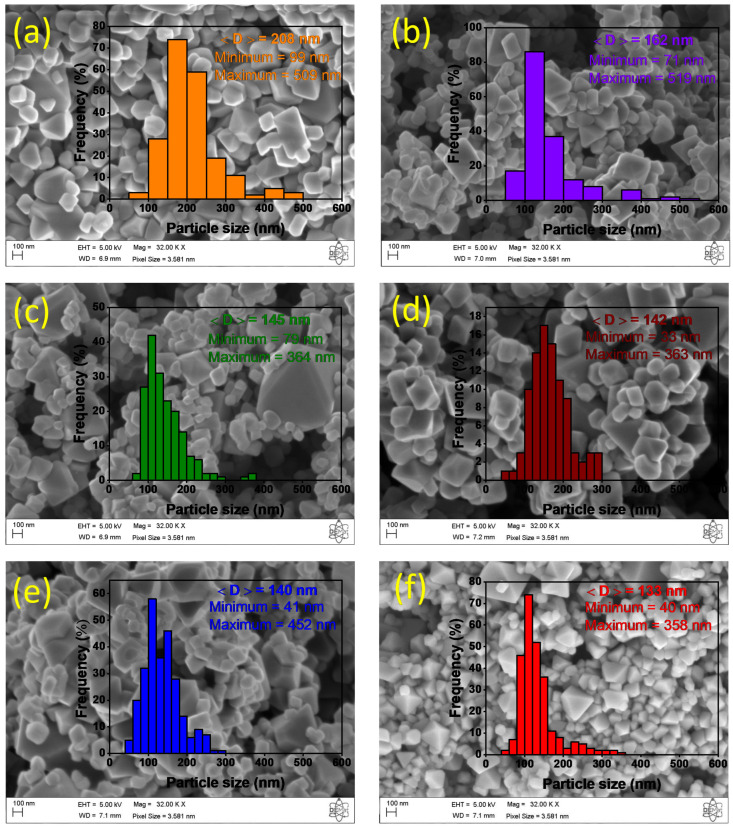
FESEM images and particle size distribution of Mn_X_Co_3-X_O_4_ (0 ≤ X ≤ 1) samples (**a**): X = 0.0, (**b**) X = 0.2, (**c**) X = 0.4, (**d**) X = 0.6, (**e**) X = 0.8, and (**f**) X = 1.0.

**Figure 5 nanomaterials-12-03170-f005:**
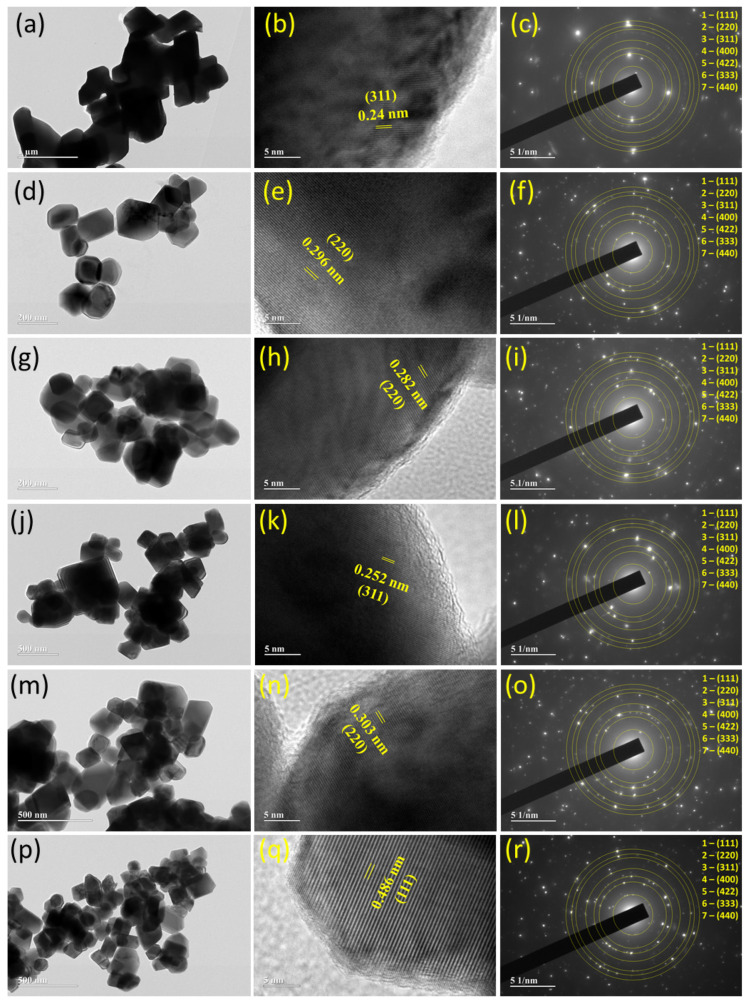
TEM images and selected area electron diffraction (SAED) of Mn_X_Co_3-X_O_4_ (0 ≤ X ≤ 1) samples: (**a**–**c**) X = 0.0, (**d**–**f**) X = 0.2, (**g**–**i**) X = 0.4, (**j**–**l**) X = 0.6, (**m**–**o**) X = 0.8, and (**p**–**r**) X = 1.0.

**Figure 6 nanomaterials-12-03170-f006:**
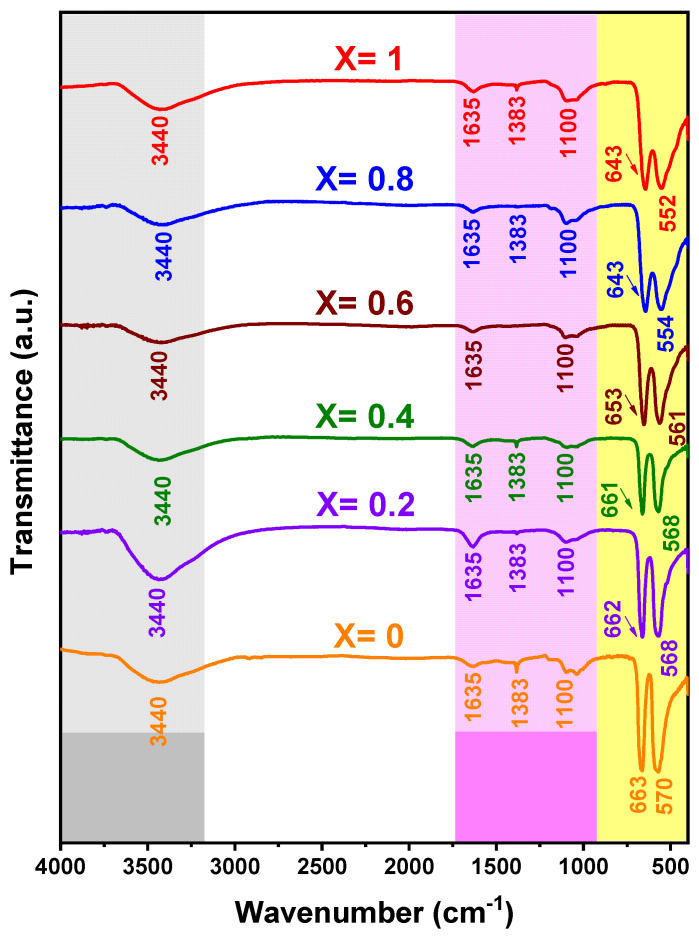
FT-IR spectra for all Mn_X_Co_3-X_O_4_ (0 ≤ X ≤ 1) samples.

**Figure 7 nanomaterials-12-03170-f007:**
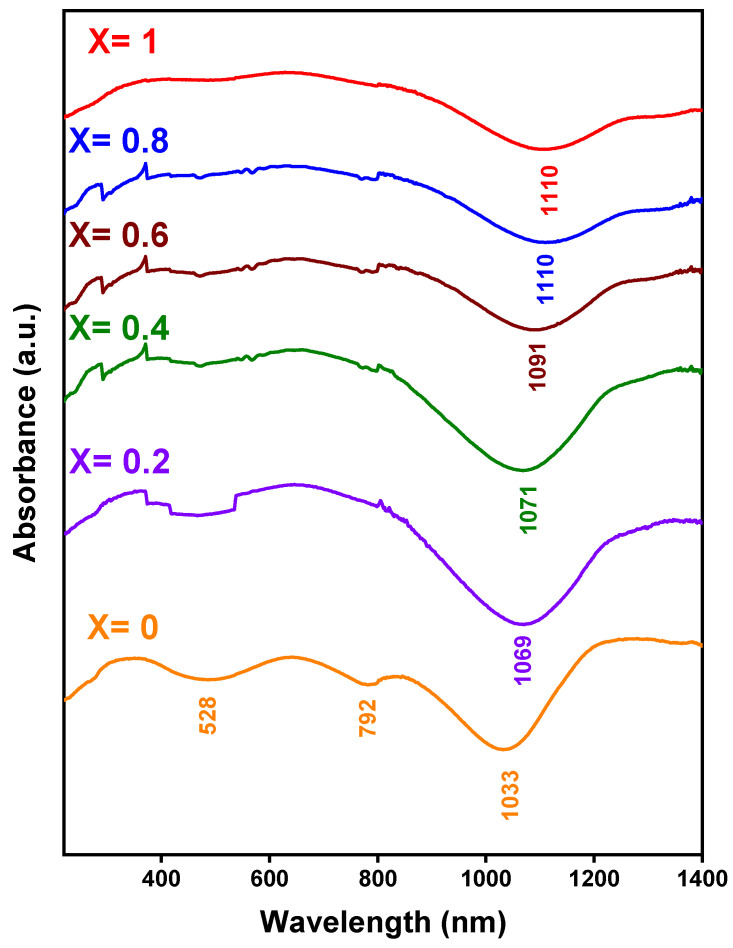
UV–Vis Absorbance spectra of Mn_X_Co_3-X_O_4_ (0 ≤ X ≤ 1) samples.

**Figure 8 nanomaterials-12-03170-f008:**
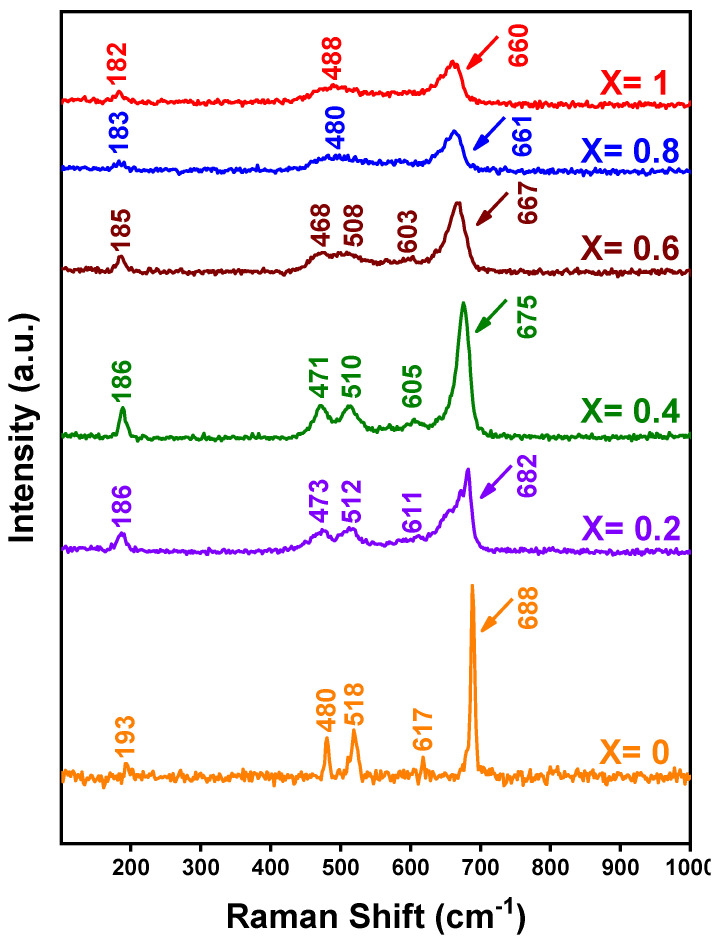
Raman spectra for all Mn_X_Co_3-X_O_4_ (0 ≤ X ≤ 1) samples.

**Figure 9 nanomaterials-12-03170-f009:**
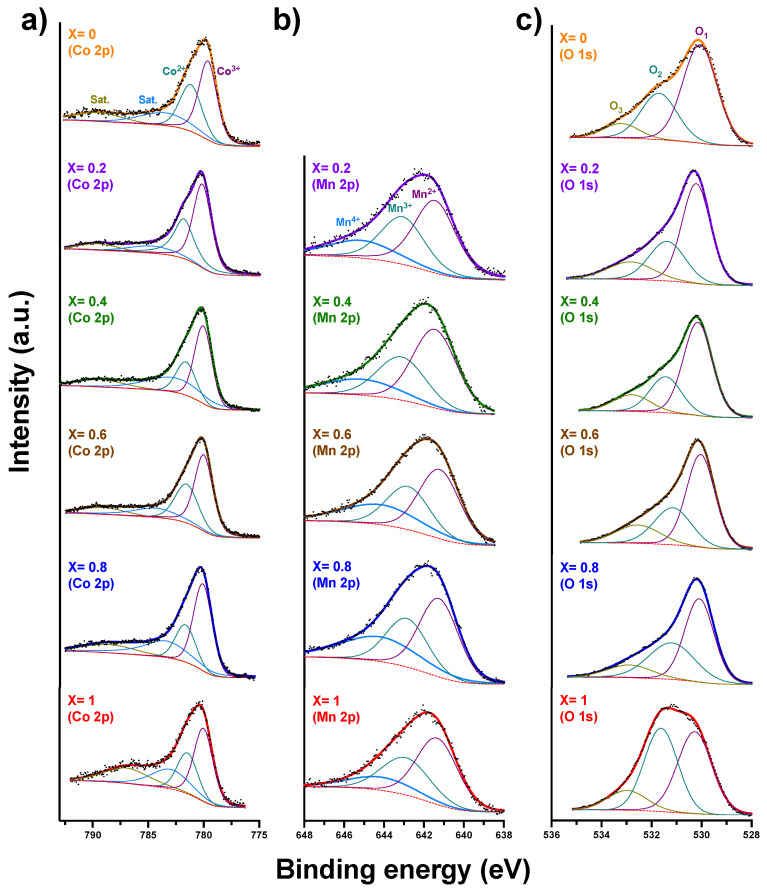
High-resolution XPS spectra of Mn_X_Co_3-X_O_4_ (0 ≤ X ≤ 1) for (**a**) Co 2p, (**b**) Mn 2p and (**c**) O 1s.

**Figure 10 nanomaterials-12-03170-f010:**
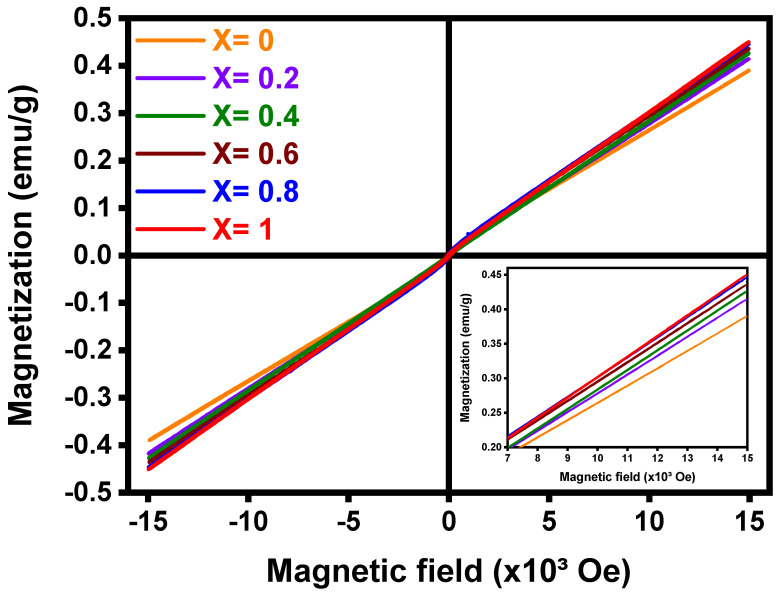
Magnetization curve of Mn_X_Co_3-X_O_4_ (0 ≤ X ≤ 1).

**Figure 11 nanomaterials-12-03170-f011:**
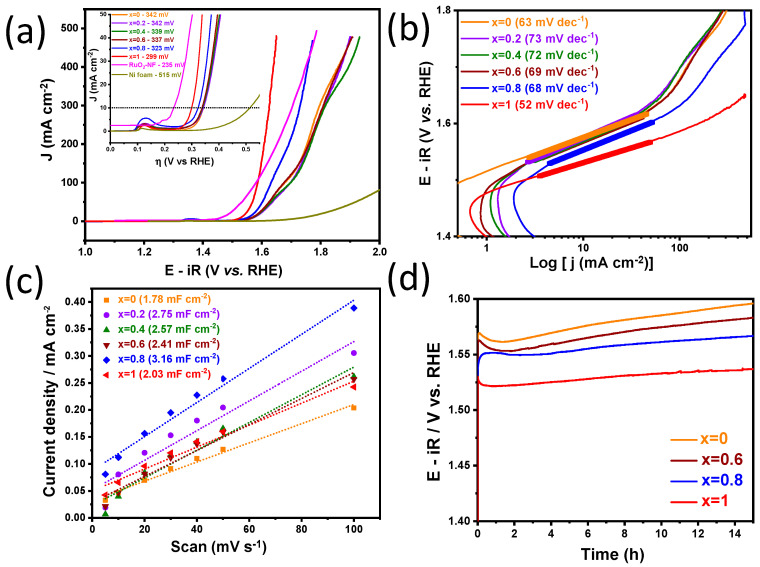
(**a**) LSV collected at 5 mV s^−1^ in 1 mol L^−1^ KOH for Mn_X_Co_3-X_O_4_ (0 ≤ X ≤ 1) electrodes, where X = 0.0, X = 0.2, X = 0.4, X = 0.6, X = 0.8, and X = 1.0, and (**b**) the corresponding Tafel slopes; (**c**) anodic current (i_a_) versus scan rate to determine CDL; (**d**) chronopotentiometry analysis measured at 10 mA cm^−2^.

**Figure 12 nanomaterials-12-03170-f012:**
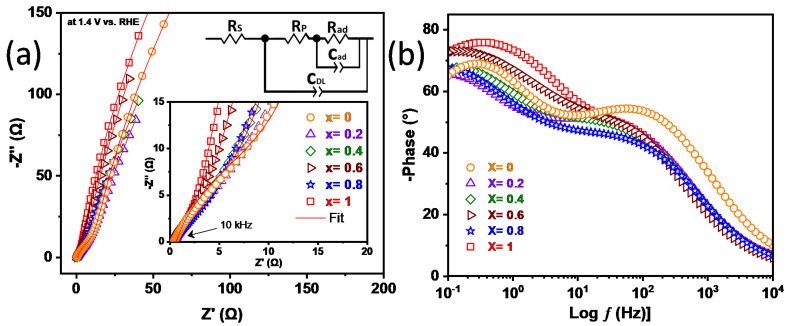
EIS—Nyquist (**a**) and Bode (**b**) plots for the tested electrocatalysts.

**Table 1 nanomaterials-12-03170-t001:** Crystallographic parameters, including crystallite size, lattice parameter, and agreement indices (*R_wp_*, *R_exp_*
*e* χ^2^) for samples of Mn_X_Co_3-X_O_4_ (0 ≤ X ≤ 1).

Samples	Co_3_O_4_—ICSD 36256		MnCo_2_O_4_—ICSD 18544		Agreement Factors		
	D_XRD_ (nm)	a (Å)	D_XRD_ (nm)	a (Å)	*R_wp_* (%)	*R_exp_* (%)	χ^2^
Co_3_O_4_—ICSD 36256	---	8.072(3)	---	---	---	---	---
MnCo_2_O_4_—ICSD 18544	---	---	---	8.28(2)	---	---	---
X = 0	90.6 **[100%]**	8.0757(1)	---	---	7.16	6.92	1.04
X = 0.2	82.5 **[100%]**	8.0759(8)	---	---	7.95	6.59	1.21
X = 0.4	80 **[100%]**	8.0754(1)	---	---	7.97	6.60	1.21
X = 0.6	71 **[100%]**	8.1183(7)	---	---	8.14	6.67	1.22
X = 0.8	68.8 **[100%]**	8.1675(8)	---	---	8.01	6.66	1.20
X = 1	---	---	66 **[100%]**	8.2381(6)	8.69	6.70	1.30

**Table 2 nanomaterials-12-03170-t002:** Assignment of the FT-IR band frequencies observed for Mn_X_Co_3-X_O_4_ (0 ≤ X ≤ 1) samples and their comparison with the literature.

Observed Frequencies (cm^−1^)							
(X = 0.0)	(X = 0.2)	(X = 0.4)	(X = 0.6)	(X = 0.8)	(X = 1.0)	Reference	Mode Assignment
3440	3440	3440	3440	3440	3440	[68,69,70,71,72]	O-H stretching vibration
1635	1635	1635	1635	1635	1635	[65,68,71,73,74]	Angular deformation of adsorbed water molecules
1383	1383	1383	1383	1383	1383	[65,66,67]	Deformations of C-N and CH_2_ groups
1100	1100	1100	1100	1100	1100	[62,68,74]	C-O stretching vibrations
663	662	661	653	643	643	[65,68,70,71,75]	Stretching vibrations of Mn–O
570	568	568	561	554	552	[21,63,64,72,73]	Stretching vibrations of Co–O

**Table 3 nanomaterials-12-03170-t003:** Raman active band positions for Mn_X_Co_3-X_O_4_ (0 ≤ X ≤ 1) samples and their comparison with the literature.

Raman Band Position (cm^−1^)						
Mn_X_Co_3-X_O_4_ (0 ≤ X ≤ 1)	F2g	Eg	F2g	F2g	A1g	Reference
(X = 0.0)	193	480	518	617	688	[82,83,84]
(X = 0.2)	186	473	512	611	682	[20,88]
(X = 0.4)	186	471	510	605	675	[81,82]
(X = 0.6)	185	468	508	603	667	[20,83]
(X = 0.8)	183	480	-	-	661	[82,89]
(X = 1.0)	182	488	-	-	660	[79,80]

**Table 4 nanomaterials-12-03170-t004:** Gaussian-fitted peaks for O 1s XPS spectra for Mn_X_Co_3-X_O_4_ (0 ≤ X ≤ 1) samples.

Sample	X = 0	X = 0.2	X = 0.4	X = 0.6	X = 0.8	X = 1
Peak (eV)	531.717	531.386	531.448	531.158	531.195	531.618
O_2_ area (nm)	9859.487	12887.39	13100.65	13363.590	15330.310	19922.580
O_2_/O_1_	0.495	0.498	0.502	0.517	0.693	0.940

**Table 5 nanomaterials-12-03170-t005:** Comparison of OER performance of nanostructured Mn_X_Co_3-X_O_4_ (0 ≤ X ≤ 1) catalysts reported in the literature. Data refer to an overpotential to generate j = 10 mA cm^2^ (ƞ_10_).

Catalyst	Substrate *	ղ_10_ (mV vs. RHE)	Tafel Slope (mV dec^−1^)	Reference
Mn_X_Co_3-X_O_4_(0 ≤ X ≤ 1) powders (ágar-ágar)	Ni foam	299	55	This work
Mn_X_Co_3-X_O_4_(0 ≤ X ≤ 2) powders	Ni foam	327	79	[16]
MnCo_2_O_4_	GC	510	123	[98]
Co_3_O_4_ nanoparticles	CFP	361	87.5	[99]
Co_3_O_4_ Nanosheet	Ni foam	190	103	[100]
Mn_X_Co_3-X_O_4_ (X = 0.3)	Ni foam	390	N.R.	[101]
Mn_X_Co_3-X_O_4_ (X = 0.6)	GC	365	50.6	[91]
MnCo_2_O_4_	Ni foam	358	N.R.	[102]
Mn_X_Co_3-X_O_4_ (1:3 ratio)	Ni foam	222	162	[103]
MnCo_2_O_4_	Ni foam	400	90	[104]
MnCo_2_O_4_	carbon cloth	400	190	[105]
MnCo_2_O_4_	GC	510	123	[98]

* CFP (carbon fiber paper); GC (glassy carbon).

**Table 6 nanomaterials-12-03170-t006:** EIS—Results of fitting of the impedance spectra reported in Figure 12.

Sample	R_S_ (Ω)	R_P_ (Ω)	C_DL_ (mF)	R_-ad_ (Ω)	C_-ad_ (mF)
X = 1.0	0.41	6.02	2.55	1292	6.07
X = 0.0	0.62	28.15	1.49	1276	3.36
X = 0.2	0.48	30.69	6.01	1917	8.47
X = 0.4	0.52	20.29	4.55	1374	8.86
X = 0.6	0.43	19.21	3.88	2149	10.34
X = 0.8	0.46	9.70	8.27	1135	11.77

## Data Availability

The study did not report any data.

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
