# Peer review of "Nanoparticles of Mixed-Valence Oxides Mn_X_CO_3-X_O_4_ (0 ≤ X ≤ 1) Obtained with Agar-Agar from Red Algae (Rhodophyta) for Oxygen Evolution Reaction"

_nanomaterials, 2022, doi:10.3390/nano12183170_

Round 1
Reviewer 1 Report
In this work, the authors developed a MnXCo3-XO4 type (0≤X≤1) mixed valence oxide by protein sol-gel method. Electrochemical testing of these samples showed that MnCo2O4 samples had excellent OER activity and long-term stability. This green synthesis method showed great potential for obtaining electrocatalysts used in oxygen evolution reaction for water splitting. I strongly recommend the acceptance of this manuscript after the following issues are addressed by the authors.
1. X in MnXCo3-XO4 is the amount of manganese salt used rather than the atomic weight of manganese in the samples. I think this expression is not rigorous, so I suggest that the author indicate the ratio of each atom in the final samples.
2. Why does the sample size decrease with the increase of Mn2+? The author should give an explanation for this phenomenon.
3. In FIG. 6, the FT-IR spectra at x=1.0 has a very small offset compared with that at x=0.8. The author should give an explanation for this phenomenon.
4. I found that the author loaded the catalyst on the nickel foam, but the author did not give the LSV of nickel foam. Therefore, I suggest that the authors give LSV of nickel foam to exclude the influence of nickel foam on the catalytic performance of the sample.
5. In this paper, with the increase of Mn2+, the catalytic performance of the samples is also improved, and the performance of the samples reaches the best when x=1.0. When x>1, can the catalytic performance be further improved? Has the author verified this?
6. Can the stability test time be extended? 15 hours is a little short.
7. The authors should give the equivalent circuit diagram for EIS fitting.
Reviewer 2 Report
In this manuscript, the author used agar as the polymerization agent for the first time and synthesized MnxCo3-xO4 oxygen evolution catalyst in a green way by changing the amount of metal nitrate. The structures of different catalysts were investigated in detail, and finally the catalysts showed significantly improved electrocatalytic performance. And the research content is very sufficient. However, this work still needs some major modifications before publication. Some specific opinions are as follows:
1. In the O 1S XPS spectrogram of MnxCo3-xO4, why the area of O2 increases a lot when x=1. At the same time, the author also lists the full width at half maxima data to further explain whether it can regulate the catalyst.
2. The author mentioned oxygen vacancies several times in his paper. Can EPR or other methods be used to further prove the existence of oxygen vacancies?
3. When preparing the working electrode, the author chooses to drop catalyst ink on the foamed nickel, whether the foamed nickel will reduce the overpotential of OER, Please add pure foamed nickel as a comparison.
4. In this work, the RuO2 data should be added to make a comparison in the OER performance test.
5. In the AC impedance test, the author chose a constant potential of 1.4 V. Whether the catalytic reaction has not occurred because the voltage is too low, and the electron transfer during the actual catalytic reaction cannot be reacted.
6. Table 4 should be presented in three-line format.
7. Some other literatures should be added. For example, Chemical Engineering Journal 431 (2022) 133829; Chem. Commun., 2022, 58, 7682–7685; Angew. Chem.Int. Ed. 2022, 61, e2021146; Journal of Alloys and Compounds 873 (2021) 159766; 10.1016/j.cej.2020.127831;
Round 2
Reviewer 2 Report
The authors have addressed the comments as possible as they can. It can be published in the present version.